# The Molecular Mechanisms of Muscle–Adipose Crosstalk: Myokines, Adipokines, Lipokines and the Mediating Role of Exosomes

**DOI:** 10.3390/cells14241954

**Published:** 2025-12-09

**Authors:** An Li, Zili Zhou, Dandan Li, Peiran Sha, Hanzhuo Hu, Yaqiu Lin, Binglin Yue, Jian Li, Yan Xiong

**Affiliations:** 1Key Laboratory of Qinghai-Tibetan Plateau Animal Genetic Resource Reservation and Utilization, Ministry of Education, Southwest Minzu University, Chengdu 610041, China; la98614@163.com (A.L.);; 2Key Laboratory of Qinghai-Tibetan Plateau Animal Genetic Resource Reservation and Utilization of Sichuan Province, Southwest Minzu University, Chengdu 610041, China; 3College of Animal & Veterinary Sciences, Southwest Minzu University, Chengdu 610041, China

**Keywords:** skeletal muscle, adipose tissue, crosstalk, cytokines, adipokines, myokines, lipokines, exosomes

## Abstract

**Highlights:**

**What are the main findings?**

**What are the implications of the main findings?**

**Abstract:**

Adipose tissue and skeletal muscle are the foremost energy depots and locomotor organs; they orchestrate metabolic homeostasis through the secretion of cytokines via autocrine, paracrine, and endocrine pathways. This intricate interplay is pivotal in the pathogenesis of numerous metabolic disorders, encompassing obesity and muscle atrophy, as well as influencing meat quality in animal production. Despite its significance, unraveling the molecular mechanisms underlying muscle–adipose crosstalk remains a major challenge. Recent advancements in multi-omics technologies have facilitated the identification of a multitude of cytokines derived from adipose tissue and muscle, including adipokines, lipokines, myokines, and myogenic exosomes and adipose-derived exosomes containing various biomolecules. The functional roles of these cytokines have been elucidated through meticulous studies employing trans-well cultures and recombinant proteins. In this comprehensive review, we summarize the bidirectional roles of adipokines and myokines in key biological processes—such as muscle satellite cell differentiation, mitochondrial thermogenesis, insulin sensitivity, and lipid metabolism. By synthesizing these findings, we aim to provide novel insights into the treatment of metabolic diseases and the improvement of animal production.

## 1. Introduction

Adipose tissue was initially identified as an energy storage organ, with excessive accumulation linked to metabolic-related diseases like type 2 diabetes mellitus (T2DM), hypertension, hyperlipidemia, cardiovascular disease, and chronic kidney disease in humans [1,2,3]. In livestock, adipose tissue deposition varies anatomically and holds different economic values and intramuscular fat deposition; in particular, their marbling score and economic worth can be enhanced by improving meat quality attributes like tenderness, flavor, and juiciness [4,5]. Over the past decades, adipose tissue has emerged as an endocrine organ, regulating metabolism through adipokines, such as leptin and adiponectin; they are predominantly secreted by adipocytes and play key roles in regulating appetite, insulin sensitivity, and inflammation [6]. For example, leptin regulates energy expenditure and food intake, while adiponectin enhances insulin sensitivity and fatty acid oxidation [7]. In terms of livestock and poultry growth, leptin concentration is positively correlated with weight gain [8,9]. In addition, leptin genotypes are closely related to the reproductive and productive traits of animals [10,11]. Dysregulation of adipokine secretion is closely associated with metabolic diseases and can also affect muscle metabolism and animal production [12,13]. Lipokines are lipid-derived signaling molecules, such as palmitoleate and 12,13-diHOME, which regulate systemic metabolic homeostasis independently of classical adipokine pathways [14]. Unlike peptide-based adipokines, lipokines are lipid metabolites that can modulate insulin sensitivity, lipid oxidation, and inflammation in distant tissues, including skeletal muscle [15]. Beyond maintaining homeostasis in energy regulation, metabolism, and cardiovascular function, adipokines exert long-term endocrine effects on metabolic status and reproduction [16]. The diverse signaling landscape of adipose tissue is further highlighted by the unique mechanisms of lipokines, which expand our understanding of their systemic influence.

Skeletal muscles account for about 40% of body weight and are crucial for motor function, respiration, and energy balance [16,17]. Similarly, muscle tissue releases various cytokines, known as “myokines”, that foster communication between muscle and other organs (such as adipose tissue or liver) [18,19]. Muscle integrity is closely related to age and metabolic health. Aging triggers coordinated dysregulation and muscle decline, manifested as intramuscular fat accumulation, insulin resistance, and mitochondrial dysfunction. Simultaneously, adipose tissue causes inflammation and ectopic fat redistribution [20]. This interaction is not only a major driver of sarcopenia, obesity, and type 2 diabetes [21,22], but also directly determines the quantity and quality of animal production [23,24]. Myokines, including myostatin, interleukin-6 (*IL-6*), and irisin, are produced and released by skeletal muscle, especially during contraction. These molecules influence muscle growth, adipose tissue browning, and systemic glucose homeostasis [25]. Myostatin, for example, negatively regulates muscle mass, and its inhibition is associated with improved lean meat yield in livestock [26]. Irisin promotes the browning of white adipose tissue and enhances energy expenditure, offering potential therapeutic avenues for obesity and metabolic disease [27,28]. In addition, irisin is closely related to animal reproduction, but further research is needed to clarify its function [29,30]. Its action highlights the diversity of adipose-derived signals.

Exosomes are small extracellular vesicles (30–150 nanometers) that can be actively secreted by almost all types of cells, including adipocytes and immune cells in adipose tissue, as well as skeletal muscle cells [31]. These tissue-specific exosomes form a sophisticated inter-organ communication network and influence processes such as satellite cell differentiation, mitochondrial thermogenesis, and lipid metabolism through their complex bioactive cargo (including proteins, lipids, various nucleic acids, and RNAs) [32]. The application of multi-omics and co-culture systems has greatly expanded the catalog of secreted factors involved in muscle–fat communication. Understanding this cytokine-mediated crosstalk—especially in the context of satellite cell differentiation, mitochondrial thermogenesis, insulin sensitivity, and lipid metabolism—will provide valuable insights for treating human adipose and muscle-related diseases, as well as for improving animal production.

## 2. The Origin of Adipocytes and Skeletal Muscle Cells

Adipocytes are usually classified into white, beige, and brown types based on their function. Anatomically, they are distributed in specific areas of the body. For example, brown adipocytes are found in the interscapular fat depot of rodents and the supraclavicular region in humans, while other major white fat depots include subcutaneous, visceral, intramuscular, and perirenal areas [33,34]. White adipocytes store excess energy in the form of triglycerides, which are utilized after fasting stimulation [35]. In fact, primary white adipocytes cultured in vitro also have a multilocular appearance. In contrast, brown adipocytes also exhibit a multilocular phenotype and express thermogenic gene profiles that uncouple mitochondrial fatty acid oxidation and ATP synthesis, enabling them to use lipids in a thermogenic way [36]. Although beige adipocytes are located within white adipose tissue, they function similarly to brown adipocytes [37]. They promote thermogenesis by upregulating uncoupling protein 1 (*UCP-1*), thereby helping to regulate body temperature homeostasis [38]. In the context of animal production, intramuscular fat cells, also known as marbling fat, are highly desirable for enhancing the flavor and palatability of meat [39]. Although all these fat cells originate from mesenchymal stem cells (MSCs) in the mesoderm, they undergo differentiation from distinct progenitors and possess specific transcriptional regulation (Figure 1).

The adipogenic process begins with the commitment of MSCs to a fat progenitor cell fate [40]. These progenitor cells then differentiate into beige or white preadipocytes under the influence of specific adipogenic factors. It is reported that MYF5^-^ progenitor cells and early adipose progenitor cells are the source of white preadipocytes [41,42]. Furthermore, MYF5^-^ progenitor cells also have the ability to differentiate into beige adipocytes [43]. Upon exposure to cold environments or after exercise, fibroblast growth factor 21 (*FGF21*) can enhance the expression of *UCP1* and *PGC-1α*, thereby promoting the conversion of white adipocytes into beige adipocytes [43,44]. Conversely, *MYF5^+^* progenitor cells produce both brown adipocytes and myoblasts; bone morphogenetic protein 7 (*BMP7*) and PR domain-containing 16 (*PRDM16*) drive the differentiation of *MYF5^+^* progenitor cells into brown preadipocytes, which eventually develop into mature brown adipocytes [45,46]. Regarding intramuscular adipocytes, non-muscle-derived mesenchymal stem cells have been identified as their primary source [47]. These cells are marked by platelet-derived growth factor receptor α (*PDGFRA*, also known as CD140a) and possess biopotency, enabling them to differentiate into either lipid-rich adipocytes or collagen I-expressing fibroblasts; they are, therefore, defined as fibro/adipogenic progenitors (FAPs) [37]. Zinc finger protein (Zfp) plays a key role in FAP’s commitment to the adipogenic lineage. It induces *PPARγ* expression, promoting the transition of FAPs to preadipocytes and their subsequent conversion into mature adipocytes [48]. Elucidating the molecular regulation of distinct adipocyte progenitors is crucial, as it will enable the targeted modulation of fat development and function.

During muscle cell formation, a normal stage of embryonic development, PAX3 and PAX7 act synergistically as myogenic factors to stimulate the differentiation of *MYF5^+^* progenitor cells into myoblasts [48,49]. Myogenic differentiation antigen 1 (*MyoD*) is a master regulator of myogenesis. It is expressed in the early stages of myogenic differentiation and induces the expression of other myogenesis-related genes. Another key regulator, myogenin (*MyoG*), promotes myoblast differentiation into myocytes [49]. Subsequently, myocytes differentiate and fuse to form myotubes [50], which ultimately mature into myofibers [50]. In the context of muscle injury, damaged muscle fibers stimulate the proliferation of muscle satellite cells, which, in turn, carry out subsequent muscle development and promote muscle regeneration [49]. By delving deeper into these comprehensive molecular mechanisms and regulatory networks, researchers will be able to develop intervention strategies to promote muscle regeneration, improve muscle function, and find treatments for related muscle diseases.

## 3. Effect of Adipose-Derived Cytokines on Skeletal Muscles

Adipose tissue releases cytokines, including adipokines, lipokines, and exosomes, maintaining individual energy homeostasis and participating in the pathogenesis of many metabolic diseases [51]. These molecules can affect adipose tissue itself, and other cells or organs (such as the heart and muscles) [52]. For skeletal muscle, these cytokines directly affect the proliferation, differentiation, and apoptosis of muscle satellite cells. In addition, adipose-derived cytokines are involved in fat ectopic deposition and insulin sensitivity of skeletal muscle, subsequently regulating its metabolism and function, and ultimately affecting human health and animal production (Figure 2).

### 3.1. Adipokines

Adipokines, a series of biologically active proteins and peptides secreted by adipocytes, are identified through various techniques, including transcriptomic and proteomic analyses. Leptin is the first adipokine to be uncovered [1] and plays an important role in communication among adipose tissue, the central nervous system (CNS), and skeletal muscle to maintain energy homeostasis [2]. In skeletal muscle, leptin increases glucose uptake and enhances insulin signaling via the activation of sympathetic nerves and β2-adrenergic receptors (β2-AR) [53]. Adiponectin, another well-known adipokine, not only improves insulin sensitivity in skeletal muscle but also exerts anti-atherosclerotic effects [54]. Interestingly, its effects appear to be context-dependent. In vitro, adiponectin induces the differentiation of goat skeletal muscle satellite cells into adipocytes [55]. Conversely, adiponectin reduces lipid content in chicken myoblasts by activating the AMPK pathway [56]. Adipocytes also secrete IL-6, which is known to induce insulin resistance and muscle mass loss in skeletal muscle [57]. Additionally, *IL-6* inhibits satellite cell proliferation and myonuclear accretion in myofibers by regulating the expression of the signal transducer and activator of transcription 3 (*STAT3*) and its target gene, cyclin D1 (*CCND1*) [58]. Studies using co-culture systems have demonstrated that adipocytes inhibit the differentiation of C2C12 myoblasts and induce the expression of *IL-6* in muscle cells [59]. This finding highlights the direct crosstalk between adipocytes and muscle cells. However, the critical role of *IL-6* in this specific process requires further confirmation.

Recently, adipokines were identified, such as Chemerin, angiopoietin-like 4 (*Angptl4*), and fibronectin type III domain-containing 4 (*FNDC4*). The exogenous addition of Chemerin promotes the proliferation of both myoblasts and smooth muscle cells, while also inhibiting myoblast differentiation through the ERK1/2 and downstream mTOR signaling pathways [60,61]. Additionally, Chemerin induces insulin resistance, mitochondrial autophagy, and dysfunction in skeletal muscle via an AKT-FoxO3a-dependent signaling pathway [62]. White adipose tissue secretes *Angptl4* in response to the systemic administration of dexamethasone (DEX), which directly stimulates cAMP-dependent PKA signaling and lipolysis [63]. In myoblasts, *Angptl4* inhibits C2C12 myogenic differentiation by suppressing the Wnt/β-catenin pathway [64]. Conversely, *FNDC4* has been identified as a novel adipokine that not only induces browning of human visceral adipose tissue but also promotes myoblast differentiation by interacting with the *LRP6* receptor to activate the Wnt/β-catenin pathway [65,66]. In bovines, *FNDC4* also promotes the migration and differentiation of skeletal muscle-derived satellite cells via focal adhesion kinase [67]. In addition, visfatin and tumor necrosis factor α (*TNFα)* derived from adipose tissue also regulate myogenesis, myocyte fusion, and muscle regeneration [68,69,70,71,72]. In summary, although these novel adipose-derived factors play crucial regulatory roles in myoblast differentiation and skeletal muscle function, direct evidence confirming that they are secreted by adipocytes to exert these specific effects is still lacking. Elucidating this direct endocrine axis represents an important direction for future research.

### 3.2. Lipokines

Bioactive lipids, referred to as “lipid factors” or “lipokines” that are produced by adipose tissue, represent a significant area of research in understanding the complex interactions between adipose and distant tissues, including muscle and liver [51]. They affect various physiological processes beyond metabolism, including immune responses and muscle function [73] (Figure 3). Lysophosphatidic acid (*LPA*) promotes the differentiation of myoblasts into myotubes and modulates muscle cell proliferation and survival [74]. Sphingosine-1-phosphate (*S1P*) induces myoblast differentiation through upregulation of gap junctional protein connexin (*Cx43*) protein expression [75]. Palmitoleic acid was previously shown to improve glucose homeostasis by reducing hepatic glucose production and enhancing insulin-stimulated glucose uptake in skeletal muscle [76], but it has also been reported to inhibit myogenic differentiation of C2C12 myoblasts through phosphorylation-dependent *MyoD* inactivation [77]. Branched fatty acid esters of hydroxy fatty acids (FAHFAs) were recently uncovered in white adipose tissues and exhibited antidiabetic and anti-inflammatory effects [78]. Melha Benlebna et al. [79] reported that eleven FAHFAs belonging to different families inhibited C2C12 myoblast proliferation, and two of the most active lipids (9-PAHPA and 9-OAHPA) induced a switch toward a more oxidative contractile phenotype of skeletal muscle in mice.

Recently, considerable attention has been given to BAT-secreted oxylipins, which form a structurally diverse family of oxidized polyunsaturated fatty acid derivatives, such as 12,13-dihydroxy-9Z-octadecenoic acid (12,13-diHOME) and 12-HEPE. Evidence showed that 12,13-diHOME treatment increased fatty acid uptake and mitochondrial fatty acid oxidation in mice skeletal muscle: a correlation that has also been observed in human vastus lateralis muscle fibers [80,81]. Similarly, 12-HEPE was demonstrated to promote glucose uptake into skeletal muscle through activation of the PI3K-mTOR-Akt-GLUT pathway [82]. Taken together, these newly discovered lipokines play a crucial role in regulating muscle function and metabolism, suggesting their potential as targets for effective interventions to improve muscle mass and treat skeletal muscle-related diseases.

### 3.3. Adipocyte-Derived Exosomes

Exosomes, small extracellular vesicles (30–150 nm) secreted by cells such as adipocytes, play a vital role in intercellular communication by transferring proteins, lipids, and RNAs [83]. Adipose-derived exosomes act as key mediators in the crosstalk between adipose tissue and skeletal muscle, influencing metabolic homeostasis and energy balance. Beyond their well-documented microRNA components, exosomes carry various protein cytokines and signaling molecules that contribute to muscle biology and systemic metabolism [84]. For instance, adipose-derived exosomes transport cytokines such as Laminins, Reelin, and PEDF, enhancing the growth of skeletal muscle [85]. Similarly, exosome-mediated delivery of HSPs and SOD2 promotes skeletal muscle generation [86].

Current research on exosome function primarily focuses on their microRNA (miRNA) components. Notably, *miR-27a* is secreted by both adipose tissue and muscle, with adipose-derived *miR-27a* specifically inhibiting insulin signaling in C2C12 cells and myocytes by downregulating *PPARγ* expression [87]. Both *miR-155* and *miR-155-5p* also originate from adipose-derived exosomes [88], and *miR-155* mimics treatment inhibits expression of the myogenic enhancer factor 2A (MEF2A) to repress myoblast differentiation [89]. Recently, Maki Itokazu and colleagues reported novel exosomes delivering *miRNA eLet-7d-3p*, which targets the transcription factor *HMGA2* to inhibit the proliferation of muscle stem cells [90]. Furthermore, gonadal white adipose tissue-derived exosomal miR-222 has been implicated in downregulating IRS1 and p-AKT levels in skeletal muscle, contributing to impaired insulin sensitivity and glucose intolerance [91]. Collectively, adipose-derived exosomes represent a crucial mechanism underlying the crosstalk between adipose tissue and skeletal muscle. Although miRNAs are prominent components involved in this process, the full spectrum of adipose-derived exosome components that influence muscle biology and metabolic health remains largely unexplored, warranting further investigation (Table 1).

## 4. Muscle-Derived Cytokine Regulation of Adipocytes

Skeletal muscle is another endocrine organ with significant secretory capabilities, producing and releasing myokines or exosomes during physical activity or in response to diverse stimuli from skeletal muscle cells (myocytes) [93,94]. These secretory molecules play pivotal roles in regulating both local and systemic metabolic processes, inflammation, muscle growth, repair, as well as interactions with adipose tissue, liver, bone, brain, and other organs [95,96]. Notably, over 100 myokines have been identified to date [97], and the secretome suggests that the skeletal muscle encompasses more than 300 secreted proteins [98]. In recent years, a surge of research has illuminated the crucial regulatory function of myokines in the crosstalk between skeletal muscle and adipose tissue. This section of the literature review will summarize the intricate interactions and regulatory mechanisms between muscle-derived secretory factors and adipose tissue, offering fresh perspectives that could revolutionize the treatment of metabolism-related diseases and enhance animal production (Figure 4).

### 4.1. Myokines

Over the past few decades, many myokines have been identified, including irisin, myostatin (*MSTN*), meteorin-like (*METRNL*), β-aminoisobutyric acid (*BAIBA*), IL-6, fibroblast growth factor 21 (FGF21), brain-derived neurotrophic factor (*BDNF*), interleukin-15 (*IL-15*), and others (Figure 3). These myokines are synthesized within muscle tissues and play pivotal roles in regulating biological processes such as lipolysis, insulin sensitivity, and glucose uptake [99,100]. Furthermore, many of them may serve as therapeutic targets for the treatment of metabolic-related diseases.

(1)
**
*Myostatin*
**


Myostatin (*MSTN*) has garnered extensive research attention across diverse species [101]. *MSTN*-knockout mice exhibit a remarkable phenotype characterized by increased muscle mass, decreased adipose tissue deposition, and a heightened resistance to obesity induced by high-fat diets or genetic mutations [102]. Further investigation has elucidated that these phenotypic changes are primarily mediated through the inhibition of *MSTN* signaling within muscle tissue, but not adipose tissue [102]. Specifically, knockdown of *MSTN* downregulates the expression of Jmjd3 via the SMAD2/SMAD3 complex, effectively inhibiting the trans-differentiation of myocytes into adipocytes [90]. Consistent with these findings, *MSTN*-knockout pigs and goats display similar phenotypic traits to those observed in mice [103,104]. Additionally, *MSTN* knockdown hinders the differentiation of adipocyte precursors into mature adipocytes by suppressing the expression of MMP-2/7/9 [101]. In vitro studies have demonstrated that direct treatment of 3T3-L1 cells with MSTN recombinant protein promotes its proliferation while inhibiting adipogenesis [105]. The reduction in fat mass following inhibition of *MSTN* is well documented, but its underlying mechanisms and specific effects on different fat depots are complex and context-dependent. For example, in *MSTN* knockout mice, fat reduction is not only associated with an increase in overall metabolic rate driven by enlarged skeletal muscle mass but also linked to direct effects on adipose tissue metabolism. Studies have shown that myostatin deficiency can promote the browning of white adipose tissue (WAT), enhancing thermogenesis and energy expenditure [106]. Although previous studies have suggested that myostatin inhibition can reduce fat, its heterogeneous effects on fat storage and metabolism warrant further investigation, particularly in the context of obesity and metabolic syndrome.

(2)
**
*Irisin*
**


*Irisin*, first identified by Pontus Boström and colleagues, emerges as a muscle-derived factor induced by exercise, possessing the capacity to induce browning of white adipocytes [27]. Treatment of preadipocytes with recombinant irisin or overexpression of *FNDC5* results in the inhibition of adipogenesis, accompanied by the downregulation of crucial adipogenic markers and activation of the Wnt pathway [107]. Irisin treatment of adipocytes enhances mitochondrial respiration and lipolysis through the PI3K-AKT signaling pathway in a time-dependent manner [108]. In human adipocytes and adipose tissue, irisin treatment upregulates the expression of thermogenesis-related genes and elevates phosphorylation levels of *p38*, *ERK*, and *STAT3* [109]. Furthermore, in vivo studies have demonstrated that sustained muscle-specific *ROCK1* activation downregulates the secretion of irisin from skeletal muscle, resulting in decreased circulating irisin levels, reduced *UCP1* expression in both brown and white adipose tissue, and impaired glucose tolerance [110]. Conversely, irisin overexpression has been shown to counteract high-fat diet-induced obesity by promoting the browning of white adipose tissue and enhancing *UCP1* expression in vivo [27]. As a dynamic myokine, the secretion of irisin is influenced by various physiological conditions. However, the question of how these fluctuations in irisin levels correlate with alterations in fat metabolism requires further exploration.

(3)
**
*Myonectin*
**


*Myonectin* (*CTRP15*), a novel member of the C1q/TNF-related protein (CTRP) family classified as a myokine, exhibits distinct expression patterns under fasting and refeeding conditions. Specifically, its levels are suppressed during fasting but undergo a significant increase following refeeding [111]. This myokine plays a pivotal role in facilitating fatty acid uptake in both adipocytes and hepatocytes [111]. Interestingly, *myonectin* inhibits adipogenesis in 3T3-L1 preadipocytes by modulating the p38 MAPK pathway and CHOP [112]. Furthermore, in porcine models, *myonectin* has been shown to inhibit intramuscular adipocyte lipid deposition by enhancing fatty acid uptake and promoting oxidative metabolism within mitochondria [113]. In vivo, *myonectin* deletion in the whole body promotes adipose fat storage and reduces liver steatosis [114]. Nevertheless, further investigation is required to ascertain whether these observed phenotypic changes are indeed attributable to myonectin originating from skeletal muscle.

(4)
**
*Metrnl*
**


Meteorin-like (*Metrnl*) is a circulating factor induced in muscle after exercise and in adipose tissue upon exposure to the cold [115]. It stimulates energy expenditure, beige fat thermogenesis, and improves glucose tolerance by increasing *IL-4* expression and promoting the alternative activation of adipose tissue macrophages [115]. In addition, administration of recombinant *Metrnl* alleviates lipid accumulation and inhibits kidney failure [116]. In humans, *Metrnl* inhibits human adipocyte differentiation by decreasing the expression of *PPARγ* and *CEBPα* [117]. Given the observed beneficial effects of *Metrnl* in experimental animals, correlation studies have been conducted to investigate the relationship between serum *Metrnl* levels and indicators such as body weight and blood lipids in both humans and animals [118,119,120]. Notably, this study revealed that *Metrnl* levels are significantly lower in obese or overweight individuals, hinting at its potential as a therapeutic target for the treatment of lipid metabolism-related diseases.

(5)
**
*BAIBA*
**


β-aminoisobutyric acid (*BAIBA*) is a small-molecule myokine [115] that induces the browning of white adipose tissue and fatty acid β-oxidation in hepatocytes both in vitro and in vivo through a *PPARα*-mediated mechanism [121]. Simultaneously, *BAIBA* reduces *LPS*-induced and high-fat diet-induced inflammation and insulin resistance in an AMPK-PPARα-dependent manner [122,123]. Clinical studies in humans have found that the serum level of *BAIBA* is inversely related to the fat mass index [124,125]: a correlation that is similar to observations in mice or rats [126]. This collective evidence indicates that *BAIBA* may be a promising therapeutic target for excessive fat accumulation.

(6)
**
*Other cytokines*
**


Notably, *IL-6* stimulates lipolysis and browning of white adipose tissue by enhancing the expression of *UCP1* [127]. In contrast, *FGF-21* is predominantly secreted by adipose tissue and skeletal muscle, with skeletal muscle contributing to lower levels of *FGF-21* expression under normal circumstances. However, muscle *FGF-21* release is elevated under conditions like fasting, exercise, and mitochondrial myopathy [128]. FGF21 promotes WAT browning by upregulating *UCP1* and *PGC-1α* expression [129]. Furthermore, *BDNF* has been shown to inhibit the differentiation of 3T3-L1 adipocytes and the browning process in differentiated adipocytes [130]. *IL-15* was previously identified as secreted by muscle, but its myogenic nature has not been reported, and its function is usually studied through the process of exogenous addition. *IL-15* enhances mitochondrial membrane potential and decreases lipid deposition in adipose tissue of mice [131], and its overexpression in skeletal muscle of mice results in reduced trunk fat mass [132]. In zebrafish fed a high-fat diet, Myricanol inhibited lipid accumulation by suppressing adipogenic factors, including *PPARγ* and *C/EBPα* [133]. Furthermore, Myricanol stimulated irisin production and secretion from myotubes to reduce lipid content in 3T3-L1 adipocytes [134].

Among the muscle-secreted factors discussed above, *myostatin* and *irisin* stand out as the two most extensively examined across different species for their regulatory roles in adipose tissue. In contrast, research on other myokines in domestic animals remains limited, particularly regarding their impact on animal production, highlighting a critical area for future studies.

### 4.2. Muscle-Derived Exosomes

Muscle-derived exosomes, small extracellular vesicles (M-EVs), serve as equally crucial intercellular messengers. They transfer bioactive cargo, including myocyte-specific proteins, lipids, and nucleic acids, to recipient cells, thereby participating in the regulation of diverse physiological and pathological processes. Acting as key vehicles in the crosstalk between skeletal muscle and other tissues—particularly adipose tissue—muscle-derived exosomes play an indispensable role in modulating systemic metabolic homeostasis, inflammatory responses, and energy balance [135,136]. For instance, skeletal muscle-secreted exosomes transport myokines—muscle-specific cytokines—such as *IL-15* and myostatin. The exosomal packaging of these factors protects them from degradation and ensures their targeted delivery to adipose tissue. Upon uptake by adipocytes, these myokines exert distinct regulatory effects on lipid metabolism, inflammatory status, and insulin sensitivity. Studies have shown that *IL-15* can stimulate the proliferation of FAPs and prevent the adipogenesis of FAPs. The growth of FAPs caused by *IL-15* was mediated through the JAK-STAT pathway [137]. Conversely, exosomal myostatin conveys a contrasting signal to adipose tissue. Studies have confirmed that myostatin delivered via exosomes suppresses adipocyte differentiation [138]. This interference contributes to adipose tissue dysfunction and is mechanistically linked to the development of obesity and associated metabolic disorders [139].

Muscle-derived exosomes carry various myokines and other signaling molecules, which can mutually influence the metabolism and function of adipose tissue [138]. Current research on M-EVs mainly focuses on miRNAs. Differential expression of miRNAs between M-EVs and adipose-derived EVs has been reported, with *miR-146a-5p* and *miR-221-5p* highly enriched in M-EVs. Further KEGG analysis revealed that these miRNAs are primarily involved in lipid-related metabolic processes, potentially through multiple signaling pathways [140]. Specifically, *miR-146a-5p* negatively regulates the *PPARγ* signaling pathway by targeting the *GDF5* gene, thereby modulating adipogenesis and fatty acid absorption [141]. During muscle regeneration, muscle stem cells (MuSCs) and their derived myoblasts/myotubes secrete extracellular vesicles enriched in *miR-206-3p* and *miR-27a/b-3p*, which repress adipogenesis in FAPs, enabling complete muscle regeneration [142]. Furthermore, the role of exosomes released by different muscle fibers in muscle–fat interactions remains to be elucidated (Table 2). An additional question is, do these exosomes also regulate lipid balance in various adipose tissue types? Therefore, a systematic investigation of the heterogeneity of M-EVs and their regulatory mechanisms on adipose tissue development is crucial for understanding the intricate balance between muscle and adipose tissue interactions.

## 5. Concluding Remarks and Future Perspectives

This review systematically delineates the intricate bidirectional communication between skeletal muscle and adipose tissue, primarily mediated by cytokines and exosomes. We consolidate evidence that adipose-derived signaling molecules, particularly those packaged within exosomes, are critical regulators of skeletal muscle metabolism, insulin sensitivity, and mitochondrial function. Conversely, we detail how muscle-derived myokines and exosomal cargo, such as miRNAs, govern adipose tissue biology by modulating lipolysis, lipogenesis, and browning. The central argument of this review is that fat–muscle interactions regulate distinct physiological processes through the secretion of various factors via different molecular mechanisms. These mechanisms may be related to human diseases (such as sarcopenia and diabetes) and the production performance of animals (such as lean meat percentage and feed conversion efficiency). The intricate interaction between muscle and adipose tissue is largely mediated by a complex network of signaling molecules, with exosomes serving as a crucial delivery system for these factors. However, several pressing challenges warrant further investigation: (1) The role of exosomes as specialized carriers in transporting cytokines and other bioactive molecules between tissues under different physiological and metabolic conditions remains to be fully elucidated. (2) Beyond proteins and miRNAs, the potential roles of other exosomal cargoes, such as lipids, circRNAs, and lncRNAs, in mediating muscle–adipose crosstalk require systematic exploration. (3) The in vivo functions and underlying molecular mechanisms of many identified muscle- and adipose-derived factors need further validation to establish their physiological relevance. (4) Critically, translating these fundamental discoveries into agricultural applications is essential. A deeper understanding of this tissue crosstalk is expected to provide novel strategies for improving livestock growth efficiency, body composition, and ultimately, animal production, thereby promoting sustainable animal agriculture.

## Figures and Tables

**Figure 1 cells-14-01954-f001:**
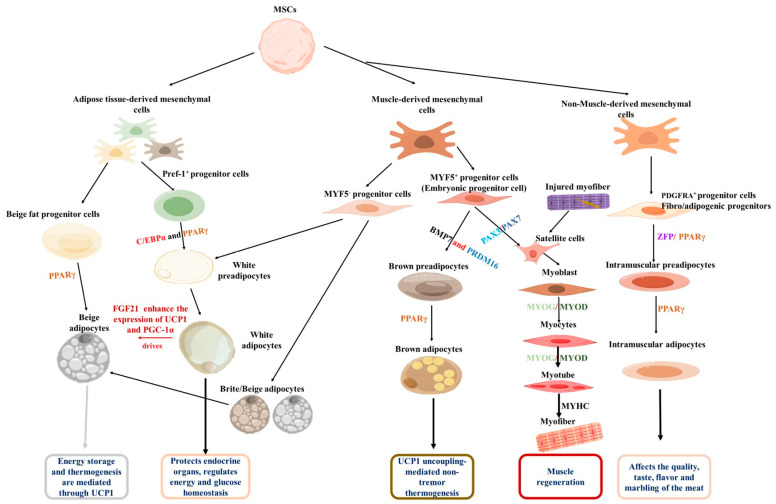
Origins of adipose tissue and skeletal muscles. Both originate from MSCs, and under the regulation of adipogenic transcription factors such as *PPARγ*, they are guided to differentiate into the adipocyte lineage, ultimately generating white, brown, and beige adipocytes, and eventually forming mature adipose tissue. Under the regulation of myogenic regulatory factors such as *Myf5* and *MyoD*, they are driven to differentiate into myogenic cells, which ultimately fuse to form multinucleated myotubes and muscle fibers. Additionally, intramuscular fat is generated under the regulation of PDGFRA.

**Figure 2 cells-14-01954-f002:**
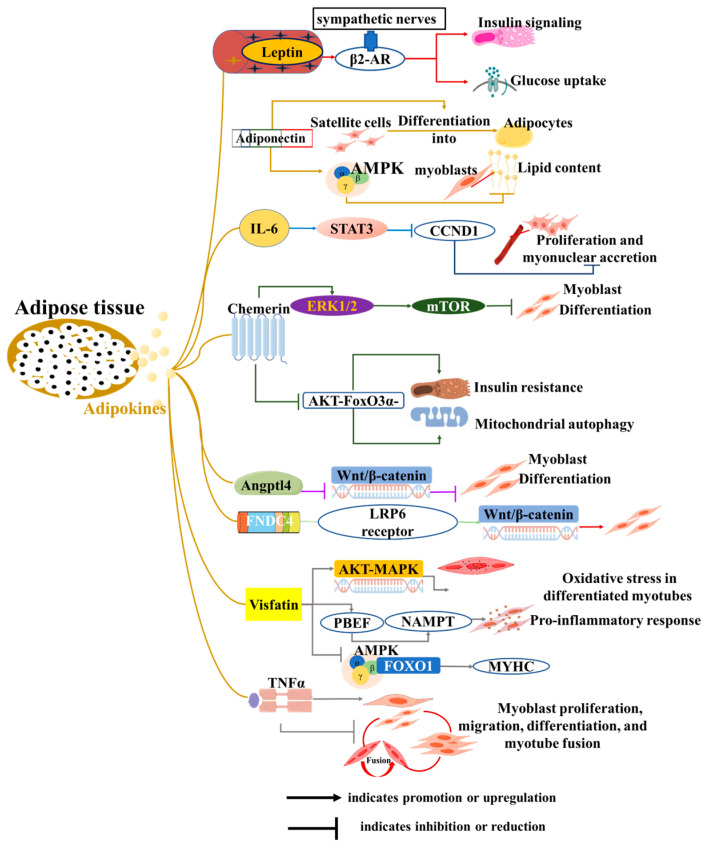
Roles of adipokines from adipose tissue to skeletal muscle development and function. The diagram illustrates how adipocytes influence various biological functions of muscle cells, including proliferation, differentiation, myotube formation and fusion, insulin resistance, and mitochondrial function, through the secretion of adipokines.

**Figure 3 cells-14-01954-f003:**
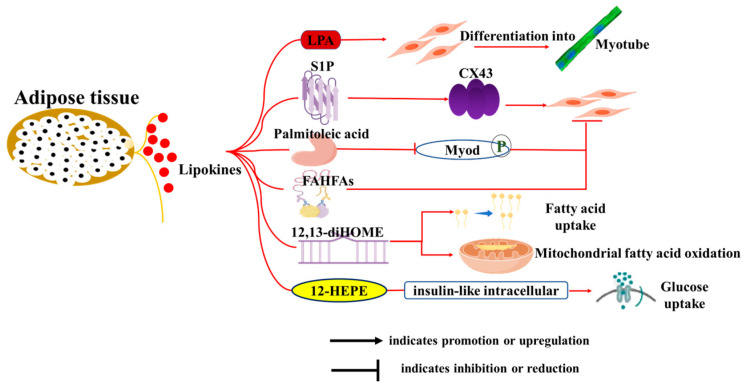
Roles of lipokines from adipose tissue on skeletal muscle development and function. The diagram illustrates how adipocytes influence various biological functions of muscle cells, including proliferation, differentiation, myotube formation and fusion, fatty acid uptake, mitochondrial function, and glucose uptake through the secretion of adipokines and lipokines.

**Figure 4 cells-14-01954-f004:**
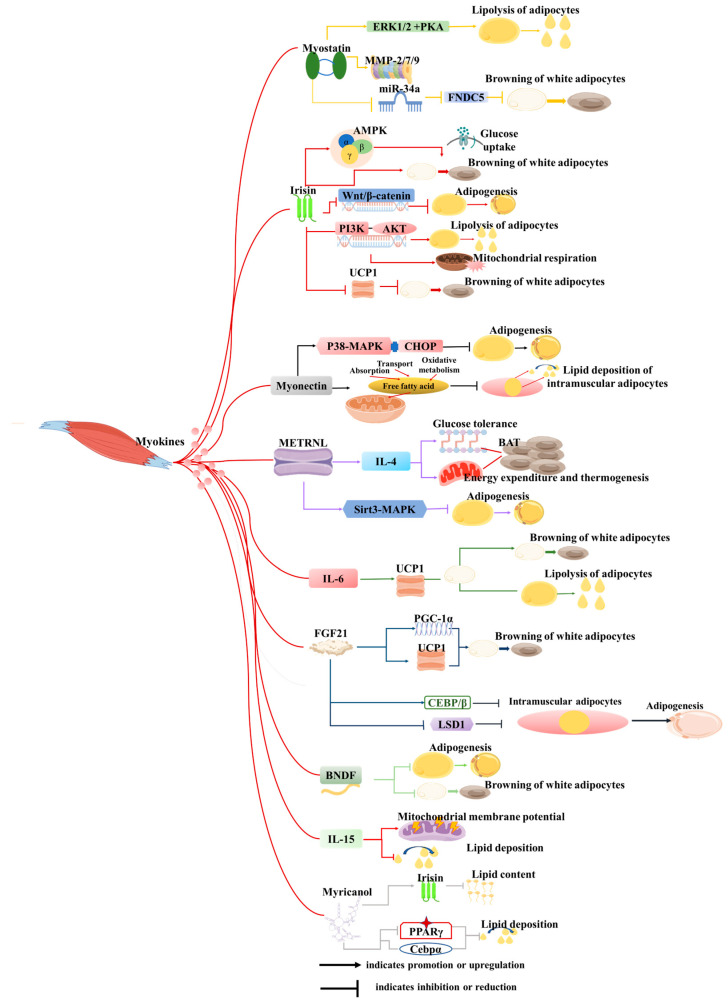
Roles of cytokines derived from skeletal muscle on adipose tissue. This figure illustrates how myokines secreted by muscles influence biological processes of adipocytes through signaling mechanisms, including proliferation, differentiation, adipogenesis, lipolysis, browning, mitochondrial function, and oxidative stress, among others.

**Table 1 cells-14-01954-t001:** Roles of exosomes from adipose tissue on skeletal muscle.

Donor	Cargoes	Recipient	Effect	Animals/ Models	Reference
Adipose tissue	Laminins, Reelin and PEDF	Muscle tissue	Enhance the growth of skeletal muscle	Mice	[85]
Adipose tissue stem cells	HSPs, SOD2	Skeletal muscle	Promote skeletal muscle generation	Mice	[86]
Adipose tissue	*miR-27a*	C2C12 cells and myocytes	Specifically inhibiting insulin signaling by downregulating *PPARγ* expression	Mice	[87]
Adipose tissue stem cells	*miR-155*	C2C12 cells	Repressing *MEF2A* expression and the inhibition of myoblast differentiation	C2C12	[89]
Adipose tissue	*m* *iR-Let-7d-3p*	Myoblast stem cells	Reducing the proliferation of myoblast stem cells by target *HMGA2*	Mice	[90]
White adipose tissue	*miR-222*	Skeletal muscle	Promote insulin resistance in the skeletal muscle by suppressing *IRS1* expression.	Mice	[91]
Preadipocytes	*m* *iR-200a*	Cardiomyocyte	Decreasing *TSC1* and subsequent mTOR activation, leading to cardiomyocyte hypertrophy.	Mice	[92]

**Table 2 cells-14-01954-t002:** Roles of exosomes from muscle on adipose tissue.

Donor	Cargoes	Recipient	Effect	Animals/Models	Reference
Muscle	*IL-15*	FAPs	Stimulate the proliferation of and prevent the adipogenesis of FAPs.	Mice	[137]
Muscle	*Myostatin*	adipocytes	Suppresses adipocyte differentiation	3T3-l1	[139]
C2C12 myoblast	*miR-146a-5p*	3T3-L1 preadipocytes	Inhibits Adipogenesis by Targeting *GDF5*-*PPARγ* Signaling	Mice	[143]
Myoblasts/Myotubes	*miR-206-3p/*	FAPs	Inhibits adipogenesis	Mice	[142]
Myoblasts/Myotubes	*miR-27a/b-3p*	FAPs	Inhibits adipogenesis by inhibiting the expression of *Pparγ*	Mice	[142]
Muscle	*miR-133*	*PRDM16*	Inhibiting the differentiation of brown adipocytes and browning of white adipocytes through inhibits the expression of *PRDM16*	Mice	[143]
Muscle	*miR-133a*	Adipocytes	Inhibits differentiation of brown preadipocytes		[144,145]
Muscle	*miR-1*	Adipocytes	Stimulates metabolic and thermogenic-related gene		[146,147]

## Data Availability

No new data were created or analyzed in this study.

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
