# Peer review of "The Molecular Mechanisms of Muscle–Adipose Crosstalk: Myokines, Adipokines, Lipokines and the Mediating Role of Exosomes"

_cells, 2025, doi:10.3390/cells14241954_

Round 1
Reviewer 1 Report
Comments and Suggestions for Authors
The study concerns the molecular mechanisms of muscle-adipose crosstalk. These issues are very interesting and still not fully explored in vivo and in vitro. The great strength of the article is the large number of references collected. Therefore, the analysis included in this article may be important for other researchers. However, the presented article requires, first of all, data organization, corrections and additions:
1/ I suggest changing the title regarding the order of parameters:
The Molecular Mechanisms of Muscle-Adipose Crosstalk: Myokines, Adipokines, Lipokines and the Mediating Role of Exosomes
2/ Graphical abstract:
The part of the figure devoted to other functions is unclear. I would prefer to focus on metabolic processes, which are the main topic of this article.
3/ Abstract:
The summary is written in a rather imprecise manner. Line 43 is unclear.
3/ Authors should avoid repetitions in the text, e.g. Lines 83-85.
4/ The description in Chapter 2 should be more compatible with Figure 1. The main factors described in this chapter should be more clearly highlighted in this figure.
5/ Figures 2 and 3 are difficult to read. I suggest omitting the graphic symbols and focusing on the diagram: parameter - metabolic pathway - metabolic effect.
6/ Some literature data are described too briefly and generally, e.g. Line 349 (reference 108).
7/ It is essential to explain all abbreviations below figures and tables. Alternatively, you could create an additional section with abbreviations and their explanations at the end of the article.
8/ Authors should pay attention to spelling in the text and tables: italics, capitalization, spaces.
9/ References:
- Line 221: The reference number should be given immediately after the author's name.
- This section concerning references should be prepared according to the journal's recommendations.
Author Response
Reviewer 1:
Comments and Suggestions for Authors
The study concerns the molecular mechanisms of muscle-adipose crosstalk. These issues are very interesting and still not fully explored in vivo and in vitro. The great strength of the article is the large number of references collected. Therefore, the analysis included in this article may be important for other researchers. However, the presented article requires, first of all, data organization, corrections and additions:
1# I suggest changing the title regarding the order of parameters:
The Molecular Mechanisms of Muscle-Adipose Crosstalk: Myokines, Adipokines, Lipokines and the Mediating Role of Exosomes
Response: Thank you for your suggestion. We have made the changes according to your recommendations.
2# Graphical abstract
The part of the figure devoted to other functions is unclear. I would prefer to focus on metabolic processes, which are the main topic of this article.
Response: We appreciate your feedback on the 'Other Functions' section in the figure. We agree that the term is ambiguous. Its original intent was to serve as a placeholder summarizing the remaining metabolic functions discussed in Figures 2 and 3. However, to prevent potential confusion and ensure that the summary figure aligns with the manuscript's primary focus on metabolism, we have modified it by removing the 'Other Functions' category.
3# Abstract:
The summary is written in a rather imprecise manner. Line 43 is unclear.
Response: Thank you for your suggestion. We have made the changes according to your recommendations. Changed these sentences into “In this comprehensive review, we summarize the bidirectional roles of adipokines and myokines in key biological processes—such as muscle satellite cell differentiation, mitochondrial thermogenesis, insulin sensitivity, and lipid metabolism. By synthesizing these findings, we aim to provide novel insights into the treatment of metabolic diseases and the improvement of animal production.” Please see the line 53-57.
3# Authors should avoid repetitions in the text, e.g. Lines 83-85.
Response: Thank you for your suggestion. We have changed these sentences into “Exosomes are small extracellular vesicles (30–150 nanometers) that can be actively secreted by almost all types of cells, including adipocytes and immune cells in adipose tissue, as well as skeletal muscle cells. These tissue-specific exosomes, form a sophisticated inter-organ communication network and influence processes such as satellite cell differentiation, mitochondrial thermogenesis, and lipid metabolism through their complex bioactive cargo (including proteins, lipids, various nucleic acids, and RNAs).” Please see the line 106-112.
4# The description in Chapter 2 should be more compatible with Figure 1. The main factors described in this chapter should be more clearly highlighted in this figure.
Response: Thank you for your suggestion. In response, we have highlighted these key PPARγ factors in the figure using different colors to fully demonstrate their crucial role.
5# Figures 2 and 3 are difficult to read. I suggest omitting the graphic symbols and focusing on the diagram: parameter - metabolic pathway - metabolic effect.
Response: We appreciate your feedback on Figures 2 and 3. To directly address the readability issues, we have standardized the colors for the regulatory pathways of key genes in the figures and indicated positive and negative regulation within the figures. Additionally, we have split Figure 2 into two separate figures for easier interpretation. Consequently, the original Figure 3 has now become the current Figure 4.
6# Some literature data are described too briefly and generally, e.g. Line 349 (reference 108).
Response: As you pointed out, some existing literature provides a high-level overview of this mechanism. We have changed this into “In humans, Metrnl inhibits adipogenesis in vitro” into “In humans, Metrnl inhibits human adipocyte differentiation by decreasing the expression of PPARγ and CEBPα.” Please see the line 387-388.
7# It is essential to explain all abbreviations below figures and tables. Alternatively, you could create an additional section with abbreviations and their explanations at the end of the article.
Response: Thank you for your suggestion. We have added the explanations for all abbreviations in the revised manuscript. Please see the line 502-529 in this revision.
8# Authors should pay attention to spelling in the text and tables: italics, capitalization, spaces.
Response: Thank you for your suggestion. We have reviewed and revised all the text based on your valuable feedback.
9# References:
Line 221: The reference number should be given immediately after the author's name.
This section concerning references should be prepared according to the journal's recommendations.
Response: Thank you for your careful review. We have made changes to the content based on this journal's recommendations. Please see the line 250.
Reviewer 2 Report
Comments and Suggestions for Authors
There are not many reviews including adipose tissue and skeletal muscle secretion (especially covering exosomes) to illuminate the cross-talk between these 2 organs. The review was original in its scope by emphasizing on applications related to human diseases and agriculture in the abstract and conclusion. But beyond the list of cytokines the review did not develop enough the rational associated with clinical application or the translation to animal production.
Previous reviews on the subjects have characterized the implication for human disease as well (Stanford 2018, Guo 2023, Fang 2023, Jia 2024, Yi 2025) or described further the significance for farm animals (Komolka 2014, Shokrollahi 2021, Shokrollahi 2024).
The direct effects on human metabolism (obesity or muscle phenotypes) could be better stated. For instance, Irisin reduces obesity (Bostrom 2012) but the authors just focused on the thermogenic aspect. There are also examples of these cytokines (such as Leptin) associated with farm animal traits (Angel 2018, Luo 2022, Yadav 2020, Montelli 2021). The authors could accentuate the applications to farm animal by citing more relevant literature, or tune down the expectation that this review is about characterizing the effect on farm animal.
The definition of “recent” or “novel” is very loose. I don’t think the discovery of BAIBA, which was published in 2014, is novel. The case could also be made for oxylipins, which were identified in 2018.
Line 69: “skeletal muscles, comprising about 40% of body weight and crucial for motor function, respiration, and energy balance“ The verb is missing.
Line 97: “Adipocytes are classified either functionally as white, beige, and brown fat cells or anatomically as subcutaneous, visceral, and intramuscular adipocytes”. This is not accurate. There are more fat depots. For instance, brown adipocytes reside in interscapular fat depot in rodent and in supraclavicular region in humans.
Line 98: reference 22 doesn’t seem perfect for the statement. There are many recent reviews describing fat depots, like Dobre 2025.
Line 100: reference 24 doesn’t seem perfect for the statement.
Line 98:” References 24,25: there are better references, such as Rosen 2014.
Line 100: “In contrast, brown adipocytes contain a large number of small lipid droplets and mitochondria, allowing them to utilize lipids in a thermogenic manner [25].” This is not accurate. Thermogenesis is associated with a specific expression program which allows mitochondrial uncoupling between fatty acid oxidation and ATP synthesis. The multilocular adipocyte phenotype are not responsible for the thermogenesis. In fact, primary white adipocyte cells have a multilocular appearance in vitro.
Line 106: reference 28 is not about “intramuscular fat cells, also known as marbling fat, are highly desirable for enhancing the flavor and palatability of meat”. It’s a review about brown fat and thermogenesis
Line 110: the sentence about Pref1 may not be important in the context of this review and the reference 29 is not about Pref1.
Line 128: “Based on the current research, the origins of various adipocyte types exhibit considerable diversity, and the further molecular regulation elucidation of different progenitors would be feasible to effectively modulate fat function and deposition.” This statement is confusing.
Line 427: “The core conclusion of this review is that the disruption of this cross-talk is a fundamental mechanism underpinning the pathogenesis of aging, sarcopenia, obesity, and type 2 diabetes, while also influencing traits of agricultural importance such as animal production”. The authors did not provide associations between the cytokines and human aging, sarcopenia or with meat production. The literature should be cited in a way with these direct effects in mind.
Figure 2: typo in Lipokines
Figure 3: typo for BAIBA. Myricanol is not described in the text
References:
Fang 2024: PMID: 36563906
Guo 2023: PMID: 37599201
Bostrom 2012: PMID: 22237023
Stanford 2018: PMID: 28507197
Jia 2024: PMID: 39741333
Yi 2025: PMID: 40957493
Komolka 214: PMID: 25057322
Shokrollahi 2021: PMID: 34175524
Shokrollahi 2024: PMID: 39109340
Angel 2018: PMID: 30343398
Luo 2022; PMID: 32787663
Yadav 2020: PMID: 32965761
Montelli 2021: PMID: 33261987
Dobre 2025: PMID: 40699742
Rosen 2014: PMID: 24439368
Author Response
Reviewer 2:
1# There are not many reviews including adipose tissue and skeletal muscle secretion (especially covering exosomes) to illuminate the cross-talk between these 2 organs. The review was original in its scope by emphasizing on applications related to human diseases and agriculture in the abstract and conclusion. But beyond the list of cytokines the review did not develop enough the rational associated with clinical application or the translation to animal production.
Response: Thank you for your comment. In response, we have added relevant content in the text regarding cytokines and their clinical applications or translation to animal production. Please see the line 73-77, line 91-96.
2# Previous reviews on the subjects have characterized the implication for human disease as well (Stanford 2018, Guo 2023, Fang 2023, Jia 2024, Yi 2025) or described further the significance for farm animals (Komolka 2014, Shokrollahi 2021, Shokrollahi 2024).
Response: Thank you for your suggestion. We add your recommended reference and other references described further the significance for human disease and farm animals. Please see the line 94, line 67, line 91, line 79, line 104, line 104, line 77, line 76, line 77, and line 76. The added reference as followed.
- Stanford, K. I.; Goodyear, L. J. Muscle-Adipose Tissue Cross Talk. Cold Spring Harbor perspectives in medicine. 2018, 8(8), a029801.20. Guo, L.; Quan, M.; Pang, W.; Yin, Y.; & Li, F. Cytokines and exosomal miRNAs in skeletal muscle-adipose crosstalk. Trends in endocrinology and metabolism: TEM, 2023;34(10), 666–681.
- Jia, Z.; Wang, Z.; Pan, H.; Zhang, J.; Wang, Q.; Zhou, C.; Liu, J. Crosstalk between fat tissue and muscle, brain, liver, and heart in obesity: cellular and molecular perspectives. European journal of medical research. 2024, 29(1): 637.
- Yi, J.; Chen, J.; Yao, X.; Zhao, Z.; Niu, X.; Li, X.; Sun, J.; Ji, Y.; Shang, T.; Gong, L.; Chen, B.; Sun, H. Myokine-mediated muscle-organ interactions: Molecular mechanisms and clinical significance. Biochemical pharmacology. 2025, 242(Pt 2), 117326.
- Komolka, K.; Albrecht, E.; Wimmers, K.; Michal, J. J.; Maak, S. Molecular heterogeneities of adipose depots - potential effects on adipose-muscle cross-talk in humans, mice and farm animals. Journal of genomics. 2014, 2, 31–44.
- Shokrollahi, B.; Shang, J. H.; Saadati, N.; Ahmad, H. I.; Yang, C. Y. Reproductive roles of novel adipokines apelin, visfatin, and irisin in farm animals. Theriogenology. 2021,172, 178–186.
- Shokrollahi, B.; Jang, S. S.; Lee, H. J.; Ahmad, H. I.; Sesay, A. R.; Ghazikhani Shad, A.; Morammazi, S.; Abdelnour, S. A. Exploring the potential roles of apelin, visfatin, and irisin in energy regulation in farm animals: an overview. Frontiers in veterinary science. 2024, 11, 1435788.
- Angel, S. P., Bagath, M., Sejian, V., Krishnan, G., & Bhatta, R. (2018). Expression patterns of candidate genes reflecting the growth performance of goats subjected to heat stress. Molecular biology reports. 2018, 45(6), 2847–2856.
- Luo, G.; Wang, L.; Hu, S.; Du, K.; Wang, J.; Lai, S. (2022). Association of leptin mRNA expression with meat quality trait in Tianfu black rabbits. Animal biotechnology. 2022, 33(3), 480–486.
- Yadav, T.; Magotra, A.; Kumar, R.; Bangar, Y. C.; Garg, A. R.; Kumar, S.; Jeet, V.; & Malik, B. S. Evaluation of candidate genotype of leptin gene associated with fertility and production traits in Hardhenu (Bos taurus × Bos indicus) cattle. Reproduction in domestic animals = Zuchthygiene, 2020, 55(12), 1698–1705.
- Montelli, N. L. L. L.; Alvarenga, T. I. R. C.; Almeida, A. K.; Alvarenga, F. A. P.; Furusho-Garcia, I. F.; Greenwood, P. L.; & Pereira, I. G. Associations of feed efficiency with circulating IGF-1 and leptin, carcass traits and meat quality of lambs. Meat science. 2021, 173, 108379.
3# The direct effects on human metabolism (obesity or muscle phenotypes) could be better stated. For instance, Irisin reduces obesity (Bostrom 2012) but the authors just focused on the thermogenic aspect. There are also examples of these cytokines (such as Leptin) associated with farm animal traits (Angel 2018, Luo 2022, Yadav 2020, Montelli 2021). The authors could accentuate the applications to farm animal by citing more relevant literature, or tune down the expectation that this review is about characterizing the effect on farm animal.
Response: Thank you for your suggestion. The related revision please refer to your comments #1 and conments#2.
4# The definition of “recent” or “novel” is very loose. I don’t think the discovery of BAIBA, which was published in 2014, is novel. The case could also be made for oxylipins, which were identified in 2018.
Response: Thank you for reviewing our manuscript. To avoid misunderstanding, “recent” or “novel” were deleted in these cases and just keep for new cytokines within five years.
5# Line 69: “skeletal muscles, comprising about 40% of body weight and crucial for motor function, respiration, and energy balance “ The verb is missing.
Response: Thank you for your comment. We have revised this sentence into “Skeletal muscle accounts for about 40% of body weight and is crucial for motor function, respiration, and energy balance”. Please see the line 88-89.
6# Line 97: “Adipocytes are classified either functionally as white, beige, and brown fat cells or anatomically as subcutaneous, visceral, and intramuscular adipocytes”. This is not accurate. There are more fat depots. For instance, brown adipocytes reside in interscapular fat depot in rodent and in supraclavicular region in humans.
Response: Your suggestion is excellent, and based on your comment, we have changed this sentence into “Adipocytes are usually classified into white, beige, and brown types based on their function. Anatomically, they are distributed in specific areas of the body. For example, brown adipocytes are found in the interscapular fat depot of rodents and the supraclavicular region in humans, while other major white fat depots include subcutaneous, visceral, intramuscular, and perirenal areas”. Please see the line 119-123.
7# Line 98: reference 22 doesn’t seem perfect for the statement. There are many recent reviews describing fat depots, like Dobre 2025.
Response: According to your suggestion, we have replaced this reference and please see the reference 33.
8# Line 100: reference 24 doesn’t seem perfect for the statement.
Response: According to your suggestion, we have replaced this reference and please see the reference 36.
9# Line 98:” References 24,25: there are better references, such as Rosen 2014.
Response: Based on your suggestion, we have replaced this reference and please see the reference 36.
10# Line 100: “In contrast, brown adipocytes contain a large number of small lipid droplets and mitochondria, allowing them to utilize lipids in a thermogenic manner [25].” This is not accurate. Thermogenesis is associated with a specific expression program which allows mitochondrial uncoupling between fatty acid oxidation and ATP synthesis. The multilocular adipocyte phenotype are not responsible for the thermogenesis. In fact, primary white adipocyte cells have a multilocular appearance in vitro.
Response: Thank you for your comment. We have revised this sentence into “In fact, primary white adipocytes cultured in vitro also display a multilocular appearance. In contrast, brown adipocytes also exhibit multilocular phenotype and express thermogenic gene profiles to uncouple mitochondrial fatty acid oxidation and ATP synthesis, enabling them to use lipids in a thermogenic way.” Please see the line 124-128.
11# Line 106: reference 28 is not about “intramuscular fat cells, also known as marbling fat, are highly desirable for enhancing the flavor and palatability of meat”. It’s a review about brown fat and thermogenesis.
Response: Thanks for your comments, replaced this reference and please see the reference 39.
12# Line 110: the sentence about Pref1 may not be important in the context of this review and the reference 29 is not about Pref1.
Response: Thanks for this comment. Given that pref-1 is not indeed as a dominant regulator during adipogenic differentiation process and, we removed the description of Pref1 and original reference 29. We have revised this sentence into “The adipogenic process begins with the commitment of MSCs to a fat progenitor cell fate [40]. These progenitor cells then differentiate into beige or white preadipocytes under the influence of specific adipogenic factors. ” Please see the line 137-139.
13# Line 128: “Based on the current research, the origins of various adipocyte types exhibit considerable diversity, and the further molecular regulation elucidation of different progenitors would be feasible to effectively modulate fat function and deposition.” This statement is confusing.
Response: Thanks for your reviewing and based on your suggestion, we have revised this sentence into “Elucidating the molecular regulation of distinct adipocyte progenitors is crucial, as it will enable the targeted modulation of fat development and function.” Please see the line 155-157.
14# Line 427: “The core conclusion of this review is that the disruption of this cross-talk is a fundamental mechanism underpinning the pathogenesis of aging, sarcopenia, obesity, and type 2 diabetes, while also influencing traits of agricultural importance such as animal production”. The authors did not provide associations between the cytokines and human aging, sarcopenia or with meat production. The literature should be cited in a way with these direct effects in mind.
Response: Thank you for your suggestion. In this regard, we have selected the key literatures you provided and added it to the text. We revised the content “Skeletal muscle abnormalities contribute to various muscle disorders such as atrophy, muscular dystrophy, and cardiomyopathy [13, 14], whose development and myofiber composition are linked to quantity and quality of animal production [15, 16].” into “Muscle integrity is closely related to age-related metabolic health. Aging triggers a coordinated dysregulation, muscle decline, manifested as intramuscular fat accumulation, insulin resistance and mitochondrial dysfunction. Simultaneously, adipose tissue occurs inflammation and ectopic fat redistribution [20]. This interaction is not only a major driver of sarcopenia, obesity, and type 2 diabetes [21,22], but also directly determines the quantity and quality of animal production [23,24]”. Please see the line 91-96. And changed “The core conclusion of this review is that the disruption of this cross-talk is a fundamental mechanism underpinning the pathogenesis of aging, sarcopenia, obesity, and type 2 diabetes, while also influencing traits of agricultural importance such as animal production. Finally, we highlight the translational potential of this knowledge, positing that tissue-specific cytokines and exosomes represent a promising new class of biomarkers for disease diagnosis and monitoring, as well as novel therapeutic targets with broad applications in both clinical medicine and animal agriculture.” into “The central argument of this review is that fat-muscle interactions regulate different physiological processes through the secretion of various factors via different molecular mechanisms. These mechanisms may be related to human diseases (such as sarcopenia and diabetes) and the production performance of animals (such as lean meat percentage and feed conversion efficiency).” Please see the line 469-473.
Figure 2: typo in Lipokines
Response: Thank you for reviewing our manuscript. We have revised this point.
Figure 3: typo for BAIBA. Myricanol is not described in the text
Response: Thank you for reviewing our manuscript. B We have revised this point. Myricanol is not described in the text. So, we have added the contentment “In zebrafish fed a high-fat diet, Myricanol inhibited lipid accumulation by suppressing adipogenic factors, including PPARγ and C/EBPα [134]. Furthermore, myricanol stimulated Irisin production and secretion from myotubes to reduce lipid content in 3T3-L1 adipocytes [135].” Please see the line 415-419.
References:
Fang 2024: PMID: 36563906
Guo 2023: PMID: 37599201
Bostrom 2012: PMID: 22237023
Stanford 2018: PMID: 28507197
Jia 2024: PMID: 39741333
Yi 2025: PMID: 40957493
Komolka 2014: PMID: 25057322
Shokrollahi 2021: PMID: 34175524
Shokrollahi 2024: PMID: 39109340
Angel 2018: PMID: 30343398
Luo 2022; PMID: 32787663
Yadav 2020: PMID: 32965761
Montelli 2021: PMID: 33261987
Dobre 2025: PMID: 40699742
Rosen 2014: PMID: 24439368
Round 2
Reviewer 1 Report
Comments and Suggestions for Authors
The Authors have corrected several aspects of the article. However, the graphical abstract does not reflect the changes made by the authors (authors' responses). Furthermore, the article still contains numerous errors and typos, including capitalization errors, e.g., lines 76,91,112.
Author Response
#1. The Authors have corrected several aspects of the article. However, the graphical abstract does not reflect the changes made by the authors (authors' responses).
Response: We agree that the term is ambiguous. To prevent potential confusion and ensure that the summary figure aligns with the main metabolic themes of the manuscript, we have revised it by removing the 'Other Functions' category from the graphical abstract.
#2. Furthermore, the article still contains numerous errors and typos, including capitalization errors, e.g., lines 76,91,112.
Response: Thank you very much for your valuable feedback and for the opportunity to improve our manuscript. We have now completed a comprehensive revision of the manuscript.